# SELF-SUPERVISED LEARNING WITH THE MATCHING GAP

## ABSTRACT

Contrastive learning (CL) is a fundamental paradigm in self-supervised learning. CL methods rely on a loss that nudges the features of various views from one image to stay closer, while pulling away those drawn from different images. Such a loss favors *invariance*: feature representations of the *same* perturbed image should collapse to the same vector, while remaining far enough from those of *any* other image. Although intuitive, CL leaves room for trivial solutions, and has a documented propensity to collapse representations for very different images. This is often mitigated by using a very large variety of augmentations. In this work, we address this tension by introducing a different loss, the *matching gap*. Given a set of $n$ images transformed in two different ways , the matching gap is the difference between the mean cost (e.g. a squared distance), in representation space, of the $n$ paired images, and the *optimal* matching cost obtained by running an optimal matching solver across these two families of $n$ images. The matching gap naturally mitigates the problem of data augmentation invariance, since it can be zero without requiring features from the same image to collapse. We implement the matching gap using the Sinkhorn algorithm and show that it can be easily differentiated using Danskin's theorem. In practice, we show that we can learn competitive features, even without extensive data augmentations: Using only cropping and flipping, we achieve 74.2% top-1 accuracy with a ViT-B/16 on ImageNet-1k, to be compared to 72.9% for I-JEPA (Assran et al., 2023).

## 1 INTRODUCTION

**Self-supervised learning (SSL).** SSL provides a principled approach, with minimal supervision, to map complex objects onto feature representations. These features have been shown to transfer well across downstream tasks with little to no fine-tuning, explaining the profound impact of SSL in computer vision (He et al., 2022; Oquab et al., 2023), natural language processing (Radford et al., 2019; Touvron et al., 2023) and speech (Baevski et al., 2020; Hsu et al., 2021). SSL methods are often described as *invariance-based methods* (Assran et al., 2023): they build upon the contrastive principle (Gutmann & Hyvärinen, 2010) that the representations for two different input objects should stay apart from each other, while making sure that each representation remains invariant to a given family of distortions of the original objects (Dosovitskiy et al., 2014; Chen et al., 2020).While simple and intuitive, the practical implementation of this idea requires a careful balancing act: Setting the magnitude and diversity of these perturbations directly impacts the trade-off between invariance and discrimination (i.e. different representations for two different inputs). When guided by performance on classification benchmarks, which are known to capture a fraction of the complexity and diversity of the visual world (Richards et al., 2023), this often leads to selecting extremely large distortions, resulting in perturbed images that are barely distinguishable from noise (Baradad Jurjo et al., 2021).

**Limitations of SSL approaches.** We conjecture that the reliance on ever-larger and diverse distortions in SSL can be interpreted as a way to mitigate the propensity of contrastive losses to force features from the same images to collapse to a single point which can be alternatively resolved using a paradigm change. A recent line of work proposes to relax the notion of invariance to equivariance (Dangovski et al., 2021; Suau et al., 2023; Gupta et al., 2023), where a small perturbation in the image corresponds to a small perturbation of the representation. However, this approach works when perturbations form a group, which is often not the case in practice. We aim for a conceptually simpler approach using ideas from optimal transport (OT) (Peyré & Cuturi, 2019).

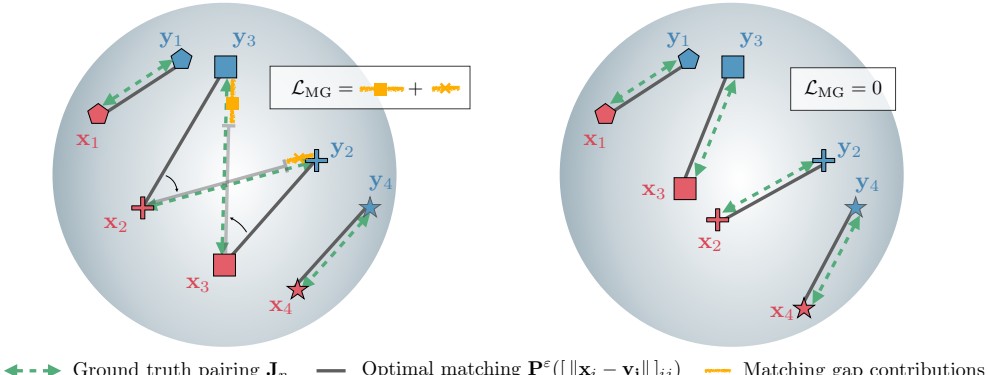

Ground truth pairing $\mathbf{J}_n$ — Optimal matching $\mathbf{P}^\varepsilon([\|\mathbf{x}_i - \mathbf{y_j}\|]_{ij})$ — Matching gap contributions

Figure 1: **Illustration of the Matching Gap**. using the Euclidean distance as a cost, the *left* figure shows a setting where the matching gap between two families of vectors is positive: the ground truth pairing has a total matching cost that exceeds that of the optimal matching. That excess is highlighted with small yellow lines, making the $\mathcal{L}_{\mathrm{MG}}$ term (6) positive. On the *right*, the ground truth pairing cost has achieved the lower bound found by the optimal matching, resulting in a 0 gap. Note that this is still possible without requiring both families of vectors to be equal.

**Our Contributions**. We propose an alternative SSL loss, the *matching gap*. This loss promotes representation maps that are content with being able to identify two representations from the same image, within a batches of $n$ images, *without* requiring them to collapse to the same point.

- The matching gap draws inspiration from (Shi et al., 2023), which linked contrastive losses to inverse optimal transport. Given a batch $(z_i)_i$ of $n$ objects, we apply two different perturbations to these objects, and then map them to vectors. This results in two paired point clouds of the same total size $n$, $(\mathbf{x}_1, \ldots, \mathbf{x}_n)$ and $(\mathbf{y}_1, \ldots, \mathbf{y}_n)$. To promote invariance, we expect that a mean cost between the $n$ coupled representations, $\frac{1}{n} \sum_{i=1}^{n} c(\mathbf{x}_i, \mathbf{y}_i)$, using a cost function $c : \mathbb{R}^d \times \mathbb{R}^d \to \mathbb{R}$, remains small. Our loss achieves contrast by subtracting the optimal matching cost, between all $(\mathbf{x}_i)_i$ and $(\mathbf{y}_j)_j$, from that mean.
- Drawing inspiration from Uscidda & Cuturi (2023) we interpret this difference as the *gap to optimality* of the ground-truth matching matrix, here the rescaled identity. As shown in Fig. 1, the matching gap between two families of vectors can be 0 even when those vectors are not equal.
- We observe that learning with a matching loss is competitive with other approaches, and works well even with no data augmentation beyond cropping. We scale up both Shi et al.'s approach and ours, the matching gap loss, to ImageNet-1k and show that they work similarly, with the matching gap being both conceptually and computationally simpler. Our approach achieves 76.7% top-1 accuracy on ImageNet-1k with a ViT-B, with augmentations, and 74.2% *without*, improving on Assran et al. (2023). This shows that our approach is a competitive alternative to the joint embedding predictive architectures paradigm (JEPA, Assran et al. (2022), LeCun (2022)).

## 2 Background: Self-Supervised Learning and Optimal Transport

### 2.1 Self-Supervised learning, with a Contrastive Lens

The aim of contrastive learning is to train a representation model that matches representations from distorted views of the same image, while pushing away views from different images. SimCLR (Chen et al., 2020) proposes to minimize a variant of the InfoNCE loss (Oord et al., 2018) between two views from images in a given batch. For each image $z_i$ from a set of $n$ images, we apply two different augmentations/perturbations $\mathcal{A}_1$ and $\mathcal{A}_2$ to obtain two images $\mathcal{A}_1(z_i)$ and $\mathcal{A}_2(z_i)$. These views are then fed to a neural network, $f_\theta$, parameterized by $\theta$, to produce $\mathbf{x}_i = f_\theta(\mathcal{A}_1(z_i))$ and $\mathbf{y}_i = f_\theta(\mathcal{A}_2(z_i))$. The InfoNCE loss between these features results in:

$$\mathcal{L}_{\mathrm{InfoNCE}}((\mathbf{x}_i)_i, (\mathbf{y}_j)_j) = -\frac{1}{n} \sum_{i=1}^{n} \left[ \mathbf{x}_i^T \mathbf{y}_i - \log \left( \sum_{j=1}^{n} \exp \mathbf{x}_i^T \mathbf{y}_j \right) \right], \tag{1}$$

This loss can be extended to subsume $\mathbf{x}_i^T \mathbf{x}_j$, $j \neq i$ into the log-sum-exp term, and is minimized w.r.t. $\theta$, the parameters of the neural network. Consider the gradient of the loss, w.r.t. one point is:

$$-\nabla_{\mathbf{x}_i} \mathcal{L}_{\text{InfoNCE}}\left((\mathbf{x}_i)_i, (\mathbf{y}_j)_j\right) = \frac{1}{n}\left[(1 - p_i(\mathbf{x}_i))\,\mathbf{y}_i - \sum_{j \neq i} p_j(\mathbf{x}_i)\,\mathbf{y}_j\right], \quad p_j(\mathbf{x}) := \frac{e^{\mathbf{x}^T \mathbf{y}_j}}{\sum_k e^{\mathbf{x}^T \mathbf{y}_k}}. \quad (2)$$

Therefore, the negative gradient w.r.t. to points of InfoNCE pushes every $\mathbf{x}_i$ towards $\mathbf{y}_i$, while pulling it away from *all* other points, in proportions that are dominated by a softmax distribution.

**Momentum Encoder.** This InfoNCE loss is computed over each batch of $n$ images and hence depends on the size of the batch. In order to simulate larger batch size, He et al. (2020) propose to compute the two feature vectors $\mathbf{x}_i$ and $\mathbf{y}_i$ using two different networks: one is called the student network, $f_{\theta_S}$ and the other has the same architecture, with parameters that are an exponential moving average (EMA) of the past values of the student network, called the teacher network, $f_{\theta_T}$. More precisely, the parameters of the teacher network are updated after seeing each batch, as $\theta_T \leftarrow \lambda \theta_T + (1 - \lambda)\theta_S$ where $\lambda \in [0, 1]$ is typically set to an initial value within the range $[0.9, 0.996]$. In this formulation, the gradient only flows through the student network, and the teacher network is considered as constant at each iteration, directly linked to the student's parameters.

**Predictive and Projecting Heads.** Another addition to the original contrastive formulation is the introduction of an auxiliary MLP head on top of these features (Chen et al., 2020). This head improves the quality of the resulting features by capturing part of the invariance promoted by the InfoNCE loss (Bordes et al., 2022). More precisely, $\mathbf{x}_i = h_{\theta_s}(f_{\theta_S}(\mathcal{A}_1(z_i)))$ and $\mathbf{y}_k = h_{\theta_T}(f_{\theta_T}(\mathcal{A}_2(z_i)))$, where $h_{\theta_s}$ is a parameterized neural network, and $h_{\theta_t}$ is parameterized with the EMA of the parameters of $h_{\theta_s}$. BYOL (Grill et al., 2020) also adds an asymmetric prediction head, $q_{\theta_s}$, to the student model alone. This setting was later adopted by Chen et al. (2021). The loss that we derive in this work enables learning self-supervised representations without a predictor network.

## 2.2 Optimal Matchings and Their Approximation with the Sinkhorn Algorithm

We take a page from the playbook of computational OT (Peyré & Cuturi, 2019), but only rely on its use in the restricted setting of optimal *matchings*, i.e. OT problems between two discrete uniform measures of the same size. Let $(\mathbf{x}_1, \ldots, \mathbf{x}_n)$ and $(\mathbf{y}_1, \ldots, \mathbf{y}_n)$ be two families of vectors in $\mathbb{R}^d$ and $c : \mathbb{R}^d \times \mathbb{R}^d \to \mathbb{R}$ a cost function between vectors. Let $\mathcal{B}_n$ the polytope of bistochastic matrices, i.e.

$$\mathcal{B}_n := \{\mathbf{P} \in \mathbb{R}_+^{n \times n} \,|\, \mathbf{P}\mathbf{1}_n = \mathbf{P}^T \mathbf{1}_n = \mathbf{1}_n/n\}.$$

Writing $\mathbf{C} = [c(\mathbf{x}_i, \mathbf{y}_j)]_{ij}$, the regularized optimal matching cost between these two families (their geometry summarized by $\mathbf{C}$), parameterized by an entropic regularization $\varepsilon \geq 0$. is defined as

$$\mathcal{S}_\varepsilon(\mathbf{C}) = \min_{\mathbf{P} \in \mathcal{B}_n} t(\mathbf{C}, \mathbf{P}), \quad \text{where} \quad t(\mathbf{C}, \mathbf{P}) := \langle \mathbf{P}, \mathbf{C} \rangle + \varepsilon \langle \mathbf{P}, \log \mathbf{P} - 1 \rangle. \quad (3)$$

When $\varepsilon = 0$, one recovers the optimal assignment problem, used for instance to compute a loss between lists of annotations in an image (Carion et al., 2020). Also known as the linear assignment problem, it can be solved with variants of the Hungarian algorithm (Kuhn, 1955) (e.g. the `scipy` routine `linear_sum_assignment`), with worst-case complexity $O(n^3)$. When $\varepsilon > 0$, the problem can be solved with the Sinkhorn fixed point iterations (Cuturi, 2013), with quadratically scaling w.r.t. sample size $n$ (Altschuler et al., 2017; Lin et al., 2019) and better GPU performance.

## 3 Contrast with The Matching Gap

Computing optimal matchings, as presented in § 2.2, is inherently an *unsupervised* task: the goal is to recover a good pairing of points (3) from two arbitrary point clouds $(\mathbf{x}_1, \ldots, \mathbf{x}_n)$ and $(\mathbf{y}_1, \ldots, \mathbf{y}_n)$, given in no specific order. Instead, in the SSL setting, data comes in the form of *pairs* $(\mathbf{x}_1, \mathbf{y}_1), \ldots, (\mathbf{x}_n, \mathbf{y}_n)$, that bind two different views of the same input. This is a *supervised* task, since the ground-truth pairing is known as the rescaled identity, $\mathbf{J}_n := \frac{1}{n}\mathbf{I}_n \in \mathcal{B}_n$.

**Optimal Transport as a Measuring Stick for Matchings.** Of what use can OT be when the ground-truth matching is known? In such cases, OT can be used as a way to quantify whether locations

$(\mathbf{x}_i, \mathbf{y}_i)$ are in agreement with the diagonal pairing: on can check whether $\mathbf{J}_n$ is similar to the best possible coupling matrix that could be computed from (3), using locations $(\mathbf{x}_i)_i$ and $(\mathbf{y}_j)_j$ to define the cost matrix. This idea was advocated in (Shi et al., 2023), who proposed to compare the ground-truth coupling $\mathbf{J}_n$, to the optimal coupling matrix to form a loss – an approach based on the idea of inverse optimal transport(Galichon & Salanié, 2022; Li et al., 2019; Stuart & Wolfram, 2020). In this work we propose a simpler approach, by focusing directly at the level of *matching costs*: we ask whether the cost of the identity ground-truth pairing, equivalent to the mean of $c(\mathbf{x}_i, \mathbf{y}_i)$, is significantly higher than the optimal matching cost that can be achieved, and use their difference as a loss. We detail first Shi et al. (2023)'s contribution, and present ours, showing that ours is easier to optimize and offers a more direct link to InfoNCE.

**Inspiration: Measuring Agreement using Inverse OT.** To turn this intuition into a quantifiable loss, (Shi et al., 2023) propose to compute a *discrepancy* between $\mathbf{J}_n$ and $\mathbf{P}^\varepsilon(\mathbf{C})$, the *optimal solution* of the matching problem parameterized by $\mathbf{C}$, the cost matrix between points $\mathbf{x}_i, \mathbf{y}_j$:

$$\mathbf{P}^\varepsilon(\mathbf{C}) := \arg \min_{\mathbf{P} \in \mathcal{B}_n} t(\mathbf{C}, \mathbf{P}), \tag{4}$$

Shi et al. (2023, Eq. 23) use then the Kullback-Leibler (KL) divergence between $\mathbf{J}_n$ and $\mathbf{P}^\varepsilon(\mathbf{C})$:

$$\mathcal{L}_{\mathrm{IOT}}(\mathbf{C}) := \mathrm{KL}(\mathbf{J}_n \| \mathbf{P}^\varepsilon(\mathbf{C})) = -\frac{1}{n} \sum_{i=1}^{n} \log[\mathbf{P}^\varepsilon(\mathbf{C})]_{ii}. \tag{5}$$

While intuitive, minimizing this loss is equivalent to a *bilevel* optimization problem: Evaluating $\mathcal{L}_{\mathrm{IOT}}(\mathbf{C})$ requires evaluating the KL divergence at the optimal solution of a lower-level optimization problem. In practice, this means being able to apply the vector-Jacobian product of $\partial \mathbf{P}^\varepsilon / \partial \mathbf{C}$. While this can be done using either unrolled Sinkhorn iterations Adams & Zemel (2011); Flamary et al. (2018) or the implicit function theorem (Luise et al., 2018; Cuturi et al., 2020; Xie et al., 2020; Eisenberger et al., 2022; Thornton & Cuturi, 2023), both approaches are computationally challenging as $n$ grows. For small values of $\varepsilon$, this requires storing a large computational graph when unrolling, or solving a linear system that is often ill-posed. For these reasons, Shi et al. (2023) rely on unrolling with a small, fixed, number of iterations (e.g. 1,2,4 or 8) regardless of $\varepsilon$. In this work, Sinkhorn is always pushed to convergence, using to a predefined threshold of $0.001$.

**Our Contribution: Single Level Optimization with the Matching Gap.** Rather than comparing $\mathbf{J}_n$ and $\mathbf{P}^\varepsilon(\mathbf{C})$ with a KL, we borrow instead an idea from (Uscidda & Cuturi, 2023), who proposed the *Monge gap* as a regularizer when carrying out the neural estimation of Monge maps (their gap was computed on the action of a neural network on a single point cloud, not paired point clouds as presented here). Transposed to our setting, their idea is to study the gap between the values of $t(\mathbf{C}, \cdot)$ evaluated at $\mathbf{J}_n$, versus $\mathbf{P}^\varepsilon(\mathbf{C})$. Simply put, we consider the *optimality gap* of the ground-truth coupling $\mathbf{J}_n$ for the matching problem:

$$\mathcal{L}_{\mathrm{MG}}(\mathbf{C}) := t(\mathbf{C}, \mathbf{J}_n) - \mathcal{S}_\varepsilon(\mathbf{C}) \tag{6}$$

$$= t(\mathbf{C}, \mathbf{J}_n) - \min_{\mathbf{P} \in \mathcal{B}_n} t(\mathbf{C}, \mathbf{P}) = t(\mathbf{C}, \mathbf{J}_n) - t(\mathbf{C}, \mathbf{P}^\varepsilon(\mathbf{C})). \tag{7}$$

$$= \frac{1}{n} \sum_i \mathbf{C}_{ii} + \varepsilon \log \frac{1}{n} - \langle \mathbf{C}, \mathbf{P}^\varepsilon(\mathbf{C}) \rangle - \varepsilon \langle \mathbf{P}^\varepsilon(\mathbf{C}), \log \mathbf{P}^\varepsilon(\mathbf{C}) \rangle. \tag{8}$$

To go from from (7) to (8), we simply use the definition of $t$, as provided in (3), which leads to a few simplifications. This definition has two benefits over (5): First, and contrary to the IOT loss, $\mathcal{L}_{\mathrm{MG}}$ can be directly differentiated using Danskin's theorem (1967) which states that it suffices when differentiating the minimum of $t$ with respect to $\mathbf{C}$, to only take into account the derivative in $t$ w.r.t. its first argument:

$$\nabla_{\mathbf{C}} t(\mathbf{C}, \mathbf{P}^\varepsilon(\mathbf{C})) = \nabla_1 t(\mathbf{C}, \mathbf{P}^\varepsilon(\mathbf{C})) = \mathbf{P}^\varepsilon(\mathbf{C}).$$

The gradient of the r.h.s. of the matching gap can be directly obtained from a forward evaluation of $\mathbf{P}^\varepsilon(\mathbf{C})$ *only*, without having to differentiate $\mathbf{P}^\varepsilon(\mathbf{C})$ w.r.t. $\mathbf{C}$ as done for IOT, to recover the simple expression $\nabla \mathcal{L}_{\mathrm{MG}}(\mathbf{C}) = \mathbf{J}_n - \mathbf{P}^\varepsilon(\mathbf{C})$. Further differentiations of $\mathbf{C}$ with respect to lower-level variables (e.g.parameters of neural networks generating locations $\mathbf{x}_i, \mathbf{y}_j$) are then simply handled using automatic reverse/mode differentiation, by prefixing these gradients with the `vjp` $(\partial \mathbf{C}/\partial \theta)^T$.

Second, the gap $t(\mathbf{C}, \mathbf{J}_n) - t(\mathbf{C}, \mathbf{P}^\varepsilon(\mathbf{C}))$ can be seen as a problem-informed way to measure the discrepancy between couplings $\mathbf{J}_n$ and $\mathbf{P}^\varepsilon(\mathbf{C})$. The gap considers matching costs, not probabilities. This is reflected in the fact that the matching gap has the following properties:

**Proposition 1.** *The Matching gap $\mathcal{L}_{\mathrm{MG}}$ as in* (6) *is a non-negative convex function on $\mathbb{R}^{n \times n}$.*

*Proof.* Non-negativity follows from the optimality of $\mathbf{P}^{\varepsilon}(\mathbf{C})$ for objective $t(\mathbf{C}, \cdot)$. Next, since $t(\mathbf{C}, \mathbf{P})$ is an affine function of $\mathbf{C}$, it is therefore concave; $\mathcal{S}_{\varepsilon}$ being the minimum over $\mathbf{P} \in \mathcal{B}_n$ of these concave functions is therefore also concave. $\mathcal{L}_{\mathrm{MG}}$, being the difference between $t(\mathbf{C}, \mathbf{J}_n)$, an affine function of $\mathbf{C}$, with $\mathcal{S}_{\varepsilon}$ a concave function, is therefore convex.

**A Link between InfoNCE and the Matching Gap.** In this section we revisit a link between OT-based losses and InfoNCE, inspired by the analysis in Shi et al. (2023, §3.2). Unlike their approach, which is limited to comparing coupling matrices, we provide an explicit relationship between the *gradients* of our matching gap loss with that of InfoNCE. Assume in this section that the ground cost $c$ is half the squared-Euclidean cost, and that all points lie in the $d$ dimensional sphere. As a result, $c(\mathbf{x}, \mathbf{y}) = 1 - \mathbf{x}^T \mathbf{y}$. We write $\mathbf{P}^{\varepsilon}$ for $\mathbf{P}^{\varepsilon}(\mathbf{C})$ to simplify notations. In that case, the derivation of the matching gap w.r.t. points $\mathbf{x}_i$ (6)can be obtained as:

$$\mathcal{L}_{\mathrm{MG}}((\mathbf{x}_i)_i, (\mathbf{y}_j)_j) = w - \frac{1}{n} \sum_i \mathbf{x}_i^T \mathbf{y}_i + \sum \mathbf{P}_{ij}^{\varepsilon} \mathbf{x}_i^T \mathbf{y}_j. \tag{9}$$

where $w = 1 - \varepsilon \left( \log n + \langle \mathbf{P}^{\varepsilon}, \log \mathbf{P}^{\varepsilon} \rangle \right)$ can be seen as a constant whose gradients vanish when taking derivatives w.r.t. $\mathbf{C}$, thanks to Danskin's theorem. As a result, one has:

$$-\nabla_{\mathbf{x}_i} \mathcal{L}_{\mathrm{MG}}((\mathbf{x}_i)_i, (\mathbf{y}_j)_j) = \frac{1}{n} \left[ (1 - \mathbf{P}_{ii}^{\varepsilon}) \, \mathbf{y}_i - \sum_{j \neq i} \mathbf{P}_{ij}^{\varepsilon} \, \mathbf{y}_j \right]. \tag{10}$$

Therefore, InfoNCE gradients 2 and the above differ mainly in the way that all contrastive points $y_j$, for $j \neq i$, are averaged: InfoNCE builds on a neighborhood-based weight vector $p(\mathbf{x}_k)$, whereas the matching gap is the result of an optimal assignment, with probabilities summarized in $\mathbf{P}_{i\cdot}^{\varepsilon}$.

**Related Works.** Some of our ideas can be traced back to the seminal Wasserstein discriminant analysis (WDA) framework proposed in (Flamary et al., 2018), which contrasts, using a ratio, a regularized Wasserstein distance between *within*-class mapped points, to that between *different*-class mapped points. While a valuable source of inspiration, (Flamary et al., 2018) restrict their attention to linear transforms, consider a handful of classes in a supervised (not self-supervised setting), and also rely on, as in (Shi et al., 2023), unrolling Sinkhorn iterations. The Sinkhorn loss $\mathcal{S}_{\varepsilon}$ also plays a role in (Chen et al., 2021) or (Guo et al., 2022), but only to *push*, in distribution, two sets of embeddings together. These references do not use $\mathcal{S}_{\varepsilon}$ to gain contrast, but only to promote overlap, as evidenced by the positive sign in front of the term $\mathcal{L}_{\mathrm{LCKT}}$ (equivalent to $\mathcal{S}_{\varepsilon}$ with our notations) in (Chen et al., 2021, Eq. 14) or in (Guo et al., 2022, Eq. 6). More recently, (Cherian & Aeron, 2020) contrast two sets of time series by *maximizing* a Wasserstein distance (here, approximated as $\mathcal{S}_{\varepsilon}$ as well) under certain transformations of the *first* set. Drawing parallels with (Paty & Cuturi, 2019), this approach does not exploit any ground-truth correspondence between the points in both sets.

In addition to the IOT approach (Shi et al., 2023) introduced above, other works (Caron et al., 2020; Oquab et al., 2023) used OT as a differentiable proxy for clustering. That is, OT was used as an intermediate step to cluster representations and not directly incorporated into the loss. Importantly, this still required differentiating with respect to Sinkhorn. Recently, (Jiang et al., 2023) proposed an OT viewpoint on SSL which builds upon the negative sample reweighting approach of (Robinson et al., 2020). Jiang et al. (2023) introduce a continuous OT formulation to sample negatives (Jiang et al., 2023, Eq.7), and obtain sample weights in practice using regularized OT (see their §7). Ultimately, their approach would yield updates that are similar to ours in (10), pending a few differences: As also adopted in (Shi et al., 2023, Alg.1), given batches of $n$ images, they rely on an $2n \times 2n$ optimal matching problem with an artificially inflated self-diagonal, to avoid self-matchings. Our approach is simpler, since it only requires matching $n$ to $n$ points, and is easier to interpret as the direct minimization of the matching gap loss.

## 4 SELF-SUPERVISED LEARNING WITH THE MATCHING GAP

Following previous SSL approaches (Balestriero et al., 2023; Grill et al., 2020) we propose minimizing the matching gap loss using a joint embedding student-teacher architecture to produce data representations. For each image, $z_i$, in a batch of $n$ images, we obtain two views of the same image

using two different (possibly random) augmentation operators $\mathcal{A}_1$ and $\mathcal{A}_2$, i.e. $\mathcal{A}_1(z_i)$ and $\mathcal{A}_2(z_i)$. These two views are then passed on as inputs to, respectively, the student network $g_{\theta_S}$ and teacher network $g_{\theta_T}$ to form two families of paired representations. We then define our training loss by instantiating the matching gap between these two batches of views (6), corresponding to:

$$\mathcal{L}(\theta_S, \theta_T) := \mathcal{L}_{\mathrm{MG}}\left([c\left(g_{\theta_S}(\mathcal{A}_1(z_i)), g_{\theta_T}(\mathcal{A}_2(z_j))\right)]_{ij}\right). \tag{11}$$

Depending on the setting, we either only compute the derivatives of $\mathcal{L}$ with respect to $\theta_S$, in which case $\theta_T$ is defined as an exponential moving average of $\theta_S$ (EMA), or update both (non-EMA).

---

**Algorithm 1** Learning a student-teach model with the matching gap : PyTorch pseudocode.

---

```
# gs, gt: student and teacher networks
# epsilon: entropic regularization
# cost_fn: (batched) OT cost function
# l: teacher momentum rate
# N: batch size
# D: feature size
gt.params = gs.params
for z in loader: # load a minibatch x with N samples
    zs, zt = augment(z), augment(z) # random views
    x = gs(zs) # student output N-by-D
    y = gt(zt) # teacher output N-by-D
    C = cost_fn(x, y) # pairwise N-by-N cost matrix between views
    loss = torch.diag(C).mean() - epsilon * torch.log(N) - W(C, epsilon)
    loss.backward() # back-propagate
    update(gs) # optimizer step on student
    gt.params = m*gt.params + (1-m)*gs.params # EMA update teacher

def W(C, epsilon): # the regularized OT cost
    with torch.no_grad():
        P = Sinkhorn(C, epsilon)
    return torch.tensordot(P, C) - epsilon * torch.entropy(P).sum()
```

---

**Architecture.** The student and teacher neural networks, $g_{\theta_S}$ and $g_{\theta_T}$ respectively, consist of a backbone $f_\theta$, a ViT-B/16 (Dosovitskiy et al., 2020), followed by a multi-layer perceptron (MLP) projection head $h_\theta$. As a result,

$$g_{\theta_S} = h_{\theta_S} \circ f_{\theta_S} , \ g_{\theta_T} = h_{\theta_T} \circ f_{\theta_T}.$$

We also consider using a predictor head, implying, $g_{\theta_S} = q_{\theta_S} \circ h_{\theta_S} \circ f_{\theta_S}$. The MLPs consist of a linear layer with output size 4096 followed by Gaussian error linear units (GeLU) (Hendrycks & Gimpel, 2016)), and a linear layer with output dimension 256.

**Augmentations.** We follow the augmentations presented in DINO (Caron et al., 2021; Grill et al., 2020). These consists of applying two different parameter sets for random color jittering, Gaussian blur and solarization of images along with multi-crop (Caron et al., 2020). We also consider a minimal augmentation setting where we only apply random resized crops, horizontal flips and normalization (see Appendix A.1).

**Training Protocol.** All models are pre-trained on the ImageNet-1k dataset (Deng et al., 2009) using self-supervision. We follow Busbridge et al. (2023) for the ViT-B/16 training procedure, for all hyper-parameters see Appendix A.1. The OT entropic regularization parameter is set to $\varepsilon = 0.5$ (we consider the impact of $\varepsilon$ on performance in Figure 2).

## 5 EXPERIMENTS

We evaluate the performance of our SSL pipeline using ImageNet-1k (Deng et al., 2009). We first evaluate the performance of a standard classification task over ImageNet, considering different architectures, as well as simplified augmentations. Next, we assess the quality of obtained features by reporting transfer learning performance over benchmarks, covering other data distributions and tasks. Finally, we evaluate the performance of the matching gap when trained over smaller datasets.

### 5.1 COMPARISONS ON IMAGENET

We compare matching gap with features extracted from alternative SSL approaches, pre-trained on ImageNet-1k in different standard settings reported in the literature.

Table 1: **Performance of SSL models pre-trained on ImageNet-1K.** Top-1 linear and $k$-NN for ViT-B/16 and ViT-L/16 architectures. For each method we present published results, methods hence differ in architecture and applied image augmentations. Total number of training epochs is reported.

| Arch. | Method | epoch | Linear | $k$-NN |
|---|---|---|---|---|
| ViT-B/16 | MoCo v3 (Chen et al., 2021) | 600 | 76.7 | - |
| | DINO (Caron et al., 2021) | 400 | 78.2 | 76.1 |
| | IBot (Zhou et al., 2021) | 1600 | 79.5 | 77.1 |
| | Matching Gap (MG, ours) | 300 | 76.7 | 72.7 |
| ViT-L/16 | MoCo v3 | 300 | 77.6 | - |
| | IBot | 1200 | 81.0 | 78.0 |
| | Matching Gap (MG, ours) | 300 | 79.2 | 74.9 |
| SOTA | Mugs (Zhou et al., 2022) | - | 82.2 | - |

**Comparing with SOTA Methods.** In Table 1, we report the best existing features pretrained on ImageNet-1k with no supervision, namely MoCo v3 (Chen et al., 2021), DINO (Caron et al., 2021) and iBot (Zhou et al., 2021). We focus on two architectures, ViT-B/16 and ViT-L/16, but also report the current best-performing features, regardless of architecture, achieved at 82.2% by Mugs (Zhou et al., 2022). Overall, we observe that our best-performing features are marginally below other SOTA methods that may leverage additional pre-training and epochs. Of particular interest, our approach is competitive with MoCo v3, which is also contrastive (see Appendix A.2 for more details).

**Comparison with the Same Number of Epochs.** Most methods increase the number of epochs during pre-training to improve their performance accuracy, usually with diminishing returns. For instance, MoCo v3 only observes a gain of 0.2% when scaling from 300 to 600 epochs. While each method may scale differently with compute, and may be more competitive after a variable number of epochs (with learning rates picked accordingly), we propose in Table 2a a snapshot picture of these performances with a budget of 300 epochs.

**Performance after Fine-tuning.** In the last set of evaluations to probe the quality of our features on ImageNet-1k, we report the fine-tuning performance. We follow the experimental setup from Chen et al. (2021), and fine-tune for 150 epochs (for further details see Appendix A.2). For completeness, we also report numbers from DeiT (Touvron et al., 2021), that was trained with supervision for 300 epochs in total. The matching gap presents competitive performance against the MoCo v3 baseline (Table 2b), while outperforming the fully supervised DeiT model.

Table 2: **Performance analysis over ImageNet-1k.** Top-1 linear performance using ViT-B/16 architecture as described in Section 4. Other results are taken from (Chen et al., 2021)

(a) **Fixed pre-training of 300 epochs.**

| Arch. | SimCLR | MoCo v3 | MG |
|---|---|---|---|
| ViT-B/16 | 73.9 | 76.5 | **76.7** |

(b) **Fine-tuning on ImageNet-1k.**

| Arch. | DeiT | MoCo v3 | MG |
|---|---|---|---|
| ViT-B/16 | 81.8 | **83.2** | 82.8 |

**Removing Augmentations.** In this experiment, we follow the setting in (Assran et al., 2023), i.e., with only cropping and horizontal flipping as data augmentations (see Appendix A.1 for details). In Table 3, we show that the relaxation introduced by the matching gap can extract enough information from only random resized crops and horizontal flips. We report top-1 accuracy for a linear probe on ImageNet-1k. Interestingly, we outperform I-JEPA, which works without augmentations, with a specific cropping setting, validating that matching pairs of views (rather than collapsing them) can help removing complex data augmentations.

Table 3: **Performance with only cropping and image flipping.** Using ViT-B/16 architecture we restrict augmentations to random cropping and flipping and report top-1 linear accuracy. Other results are taken from (Assran et al., 2023).

| Method | Arch. | #epoch | Linear |
|---|---|---|---|
| MAE | ViT-B/16 | 1600 | 68.0 |
| I-JEPA | ViT-B/16 | 600 | 72.9 |
| Matching Gap | ViT-B/16 | 300 | **74.2** |

## 5.2 TRANSFER LEARNING

The quality of features pre-trained on ImageNet-1k can be evaluated by testing performance on benchmarks covering other data distributions. We follow the common setup where features are fine-tuned on each benchmark (Chen et al., 2021; Caron et al., 2021), and consider seven benchmarks: CIFAR-10, CIFAR-100 (Krizhevsky et al., 2009), Oxford Flowers-102 (Nilsback & Zisserman, 2008), Oxford-IIIT-Pets (Parkhi et al., 2012), Cars (Krause et al., 2013), iNaturalist2018 (Van Horn et al., 2018), and iNaturalist2019 (Van Horn et al., 2019). In Table 4, we report the top-1 performance on the different benchmarks (for details see Appendix A.2).

Table 4: **Transfer learning performance.** We fine-tune features pre-trained on ImageNet-1k over downstream tasks and report top-1 linear accuracy. Numbers for other methods are directly quoted.

| Method | CIFAR-10 | CIFAR-100 | Flowers | Pets | Cars | iNat$_{18}$ | iNat$_{19}$ |
|---|---|---|---|---|---|---|---|
| MoCo v3 | 98.9 | 90.5 | 97.7 | 93.2 | - | - | - |
| DINO | **99.1** | **91.7** | **98.8** | - | 93.0 | 72.6 | 78.6 |
| Matching Gap | **99.1** | 91.6 | 97.0 | **94.0** | **93.3** | **75.4** | **79.3** |

## 6 ABLATION STUDIES

In this section we study the dependence on architecture choices and hyperparemeters of the matching gap objective. We study ablations of heads and EMA, as well as the impact of entropic regularizer $\varepsilon$ on performance. Additional ablation studies, including robustness to pre-training on small datasets and further comparisons to (Shi et al., 2023), are provided in Appendix A.3.

**Robustness to Different Heads.** The common practice in SSL is to add an MLP head, called the projector, on top of the backbone encoder (Chen et al., 2020), or even, two different heads, a projector and a predictor (Grill et al., 2020). Bordes et al. (2022); Appalaraju et al. (2020) have recently explained the role of these heads as being able to account for invariances induced by data augmentation. Since the matching gap loss does not enforce strongly these invariances, we test whether our approach is robust to removals of these heads. We train a ViT-B/16 for 300 epochs with both projection and prediction heads (Proj.+Pred.) or only with a projection head (Proj.). In Table 5a, we show that the matching gap obtains better performance without the predictor head. In comparison, BYOL (Grill et al., 2020), an alternative to contrastive learning, does not converge without the predictor head.

Table 5: **Ablation studies for model components.** We test model robustness to removal of architecture and training components. Results for MoCo v3 are from Chen et al. (2021).

(a) **Performance with different heads.**

| ViT-B/16, 300 ep. | Proj.+Pred. | Proj. |
|---|---|---|
| MoCo v3 | **76.5** | 75.5 |
| Matching Gap | 76.4 | **76.7** |

(b) **Performance without EMA.**

| ViT-B/16, 300 ep. | without EMA | with EMA |
|---|---|---|
| MoCo v3 | 74.3 | 76.5 |
| Matching Gap | **75.0** | **76.7** |

**Removing the Momentum Encoder.** We set the momentum coefficient ($m$) to zero, implying that the teacher is updated simultaneously with the student and evaluate the performance. As observed by MoCo v3 Chen et al. (2021), this is expected to showcase a drop in performance. Notably, assessing the performance of matching gap without the momentum encoder (without EMA in Table 5b), we find that matching gap is relatively robust to this removal and retains good performance.

**Features of Optimal Transport Objectives.** The entropic regularization parameter, $\varepsilon$, plays a central role in obtaining the OT solution, and we expect it to impact overall model performance. To assess this, we train and evaluate matching gap models, as well as IOT models (Shi et al., 2023), over a range of $\varepsilon$. In doing so, we test for the first time the IOT objective on ImageNet-1k. In Figure 2, we report the top-1 $k$-NN and linear accuracy of both models. We empirically see that value $\varepsilon = 0.5$ performs the best (recall that the values of our cost, being equal to the squared Euclidean distance between points of the sphere, ranges from 0 to 4). These experiments confirm that a medium $\varepsilon$ value works well: From (10), a very small $\varepsilon$ would result in a loss that only pushes a point $\mathbf{x}_i$ away from

its optimal match $\mathbf{y}_{j^\star}$, and closer to its ground truth pairing $\mathbf{y}_i$. Conversely, setting $\varepsilon$ to a very large value would result in a naive gradient pressure moving the point $\mathbf{x}_i$ away from the naive mean of all $\mathbf{y}_j$, since all coefficients $\mathbf{P}_{ij}^\infty$ would collapses to $1/n^2$. A reasonably small $\varepsilon$ computes weights adaptively and mitigates the pressure to obtain invariance without being dominated by it. As seen by our results, both IOT and the matching gap perform similarly (with the latter having a negligible edge for the best $\varepsilon$ setting; Matching Gap=76.7%, IOT=76.5%), confirming that they do capture in their loss similar information. We believe the matching gap remains more promising in practice, because of its simpler interpretation and computation (only a forward pass for Sinkhorn, on a $n \times n$ problem, versus a backward Sinkhorn on $2n \times 2n$ points, with a diagonal penalty).

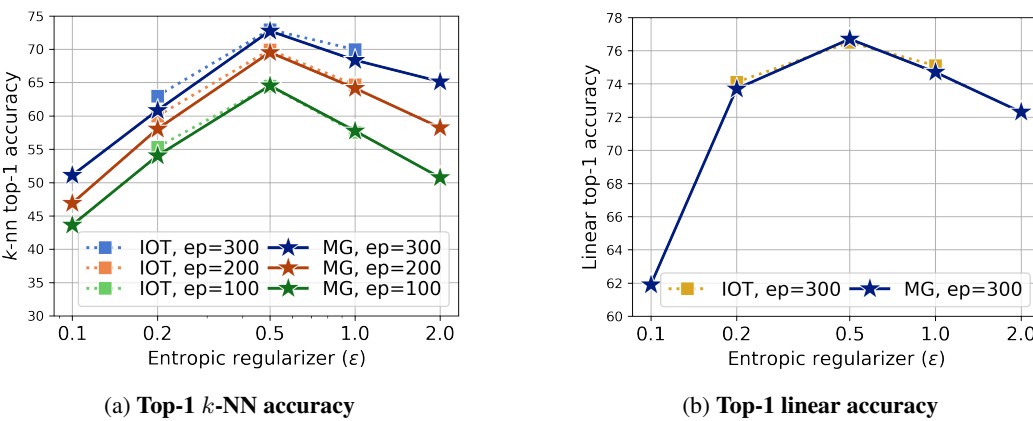

(a) **Top-1 $k$-NN accuracy**       (b) **Top-1 linear accuracy**

Figure 2: **Top-1 performance as a function of the entropic regularizer** $\varepsilon$. We test the (a) $k$-NN and (a) linear accuracy of matching gap (MG, $\star$) as well as our implementation of IOT ($\square$) as a function of the regularization parameter $\varepsilon$. $k$-NN results are reported every 100 epoch intervals, whereas linear evaluation is only performed after the last epoch (300). For $\varepsilon = 0.5$ the Linear top-1 accuracy for the matching gap is 76.7% and for IOT it is 76.5%. For both methods, the number of Sinkhorn iterations scales with $\varepsilon$, with, e.g., a mean of 90 iterations for $\varepsilon = 0.1$, 40 for $\varepsilon = 0.2$ and up to 10 iterations for $\varepsilon \geq 0.5$ and above. Unlike the approach advocated in (Shi et al., 2023), we let Sinkhorn run to convergence at any of these $\varepsilon$ levels (we were not able to find their $\varepsilon$ choice) and unroll through these iterations. We only tested IOT for $\varepsilon \in \{0.2, 0.5, 1.0\}$.

LIMITATIONS

Our approach relies on the Sinkhorn algorithm, and requires tuning $\varepsilon$. In our experiments, this did not play an important role, since we quickly narrowed down on a range for $\varepsilon$ that only required a few dozens iterations, with a small mini-batch size of $n = 128$. If one were to scale up to much larger batch sizes, the quadratic complexity of Sinkhorn could be a limiting factor when $n$ is a few thousands. In that regime, the matching gap could benefit from alternative low-rank OT solvers (Scetbon et al., 2021; Scetbon & Cuturi, 2022) that have linear scaling with $n$. This solution would work with the matching gap, but not the IOT approach, because such low-rank solutions cannot be differentiated efficiently.

CONCLUSION

The matching gap is a loss for SSL, designed for a joint embedding architecture. This loss is computed on a batch of $n$ objects, transformed according to two different views. The cost or distance between the representations of the two views of the same object is first averaged over the batch. This value, which should be ideally small to encourage invariance, is modified by removing the optimal matching cost taken between all the aggregate first and second views. When trained for a short number of epochs, an SSL architecture that minimizes the matching gap obtains comparable results to state-of-the-art frameworks on standard classification tasks, as well as in transfer learning to different datasets. This performance is only slightly impacted when removing complex data augmentations. Lastly, the matching gap loss does not require asymmetry in the model architecture, and retains good performance after removal of the momentum encoder.

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

## A   APPENDIX

### A.1   IMPLEMENTATION DETAILS

**Sinkhorn Optimization.**   To perform experiments we implemented the Sinkhorn algorithm in Py-Torch Paszke et al. (2019). We used the same implementation for both the matching gap loss and the IOT (Shi et al., 2023) objective, scaling the latter to ViT architectures (Dosovitskiy et al., 2020) and the ImageNet-1k dataset (Deng et al., 2009). The algorithm is always run to convergence, to the extent that the deviation of both marginals of the coupling variable to the targeted marginal $\mathbf{1}_n/n$ must be lower, in 1-norm, to $0.001$. For IOT, we differentiate its output using unrolling.

**Hyperparameters for Trained Models**   In Table A1 we provide the hyperparameters used to train the matching gap on ViT-B/16. For ViT-L/16 we modify the mini batch size to 32 and following (Busbridge et al., 2023) we scale the base teacher momentum to 0.997.

**Augmentations.**   We follow the augmentation stack presented in  Caron et al. (2021); Grill et al. (2020). Image transformations include color jittering, Gaussian blur and solarization. For each image we provide two views and ten multi-crops. In the no augmentation regime we avoid any color manipulation. In practice, we use the standard augmentations use in linear evaluation, implying random horizontal flipping, random resized cropping and normalization.

### A.2   EVALUATION

**Linear Evaluation.**   We follow the common practice for linear evaluation  Chen et al. (2021); He et al. (2016). Following pre-training we obtain frozen features from the backbone and train a linear classifier over ImageNet-1k for 100 epochs in a supervised manner. We use the SGD optimizer, with a batch size of 4096, sweep over $learning\_rate$ values and with zero $weight\_decay$. For image augmentations we use only random horizontal flipping, random resized cropping and normalization.

$k$**-NN Classification.**   We follow standard practice, fixing the number of neighbors to $k = 20$, and setting the voting temperature as $T = 0.07$.

**Transfer Learning.**   We follow the procedure reported in DINO Carion et al. (2020). We fine-tune the model for the downstream task. In accordance with DINO, we fine-tune CIFAR-10 [1], CIFAR-100[2] and Flowers datasets for 1000 epochs, $\text{iNat}_{18}$[3] and $\text{iNat}_{19}$ for 300 epochs. Pets and Cars datasets are fine-tuned for 100 epochs. Following this, we use the same hyperparameters and augmentations stacks.

**End-to-end Fine-tuning.**   We follow the end-to-end fine-tuning procedure reported in  Touvron et al. (2021). We replace the heads with a linear classifier and train the full model in a super-vised manner for 150 epochs over ImageNet-1k. We use AdamW optimizer, a batch size of 4096, $learning\_rate$ of $5 \times 10^{-4}$ with 20 warm-up steps and a cosine scheduler. Weight decay is kept constant at $0.05$.

---

[1] https://github.com/facebookresearch/dino/issues/81

[2] https://github.com/facebookresearch/dino/issues/144

[3] https://github.com/facebookresearch/dino/issues/171

**Training and Evaluation on Smaller Datasets.** For the smaller datasets, CIFAR-10, CIFAR-100 and SVHN we follow the pre-training procedure described in Shi et al. (2023). We use ResNet-50 as a backbone, followed by a single MLP projection head. We train for 500 epochs with a mini-batch size of 128. We use the AdamW optimizer with a $learning\_rate$ of $3 \times 10^{-4}$ and constant $weight\_decay$ of $1 \times 10^{-6}$. For augmentation we use the standard augmentation stack of random color jitter, grayscale, horizontal flip and cropping. Following pre-training, for linear evaluation we freeze the backbone and train a linear classifier for 100 epochs using an SGD optimizer.

Table A1: matching gap **ViT-B/16 hyperparameters.**

|  | BYOL ViT-B/16 |
|---|---|
| ImageNet 1k Linear Probe Test Top-1 | 76.7% |
| Weight initialization | `trunc_normal(.02)` |
| Backbone normalization | LayerNorm |
| Batch size | 4096 |
| Head normalization | LayerNorm |
| Synchronized BatchNorm over replicas | True |
| Learning rate schedule | Single Cycle Cosine |
| Learning rate warmup (epochs) | 10 |
| Learning rate minimum value | $5 \times 10^{-5}$ |
| Training duration (epochs) | 300 |
| Optimizer | AdamW |
| Optimizer scaling rule | Adam |
| Base $(\beta_1, \beta_2)$ | (0.9, 0.95) |
| Base learning rate | $6.5 \times 10^{-4}$ |
| OT Mini batch size | 128 |
| Base teacher momentum | 0.99 |
| Weight decay | 0.04 |
| Weight decay end | 0.4 |
| Weight decay warmup | 0 |
| Augmentation stack | DINO (Caron et al., 2021) |

## A.3 ADDITIONAL ABLATION STUDIES

**Robustness to Small Pre-training Data.** Contrastive learning methods require large pre-training dataset to work, as shown in El-Nouby et al. (2021). In the small data regime, the task of contrasting between images is too simple and they overfit. A similar observation is made in Shi et al. (2023) where they also propose a optimal transport based approach as an alternative. In this set of experiments, we probe if our approach is a better alternative in the low data regime to both contrastive methods and the method proposed by Shi et al. (2023), denoted IOT-CL-uniform. We follow their experimental setting and train a ResNet-50 He et al. (2016) backbone with matching gap on smaller datasets, that are CIFAR-10, CIFAR-100 Krizhevsky et al. (2009) and SVHN Netzer et al. (2011) (for further details see Appendix A.2). In Table A2, we show that the matching gap outperforms both IOT-CL-uniform and SimCLR. While our approach is slight better than IOT-CL-uniform, we are still far from the performance of a network train end-to-end with supervision on these datasets. This indicates that optimal transport based solution is still suboptimal in the low data regime.

Table A2: **Pre-training on small datasets.** We pre-train a ResNet-50 backbone over each dataset, CIFAR-10, CIFAR-100 and SVHN, and report top-1 linear performance and $k$-NN evaluation for $k = 5$. Results for other methods are taken from Shi et al. (2023).

|  | CIFAR-10 | | CIFAR-100 | | SVHN | |
|---|---|---|---|---|---|---|
|  | linear | $k$-NN | linear | $k$-NN | linear | $k$-NN |
| SimCLR | 90.6 | 86.8 | 66.3 | 53.0 | 91.6 | 75.3 |
| IOT | 91.0 | **87.6** | 67.6 | 55.7 | 93.2 | 84.1 |
| Matching Gap | **91.5** | 86.9 | **69.4** | **56.2** | **94.3** | **89.3** |

