# OpenReview forum: "Self-Supervised Learning with the Matching Gap"
_ICLR.cc/2024/Conference — Submitted to ICLR 2024_

### Official Review · Reviewer_PMfK · 2023-10-31

**Soundness:** 3 good
**Presentation:** 3 good
**Contribution:** 3 good
**Rating:** 5
**Confidence:** 4

**Summary:**

The paper presents a new self-supervised method that reduces the gap between the ground-truth matching and optimal matching. The proposed loss, matching gap, could potentially alleviate the problematic signal that force different views of the same image to collapse to the same point, even when they capture dramatically different contents, i.e. foreground vs background. To further ease the optimization, the paper proposes to learn the optimal matching via the Sinkhorn algorithm. Experimental results show that the proposed method performs on par with SOTA approaches with strong data augmentations and outperforms several latest self-supervised works with simpler data augmentations.

***
Post-rebuttal comment:

Following the discussion with the authors regarding their revision, the reviewer has maintained the original rating (marginally below the acceptance threshold), as the revision has not fully addressed the reviewer’s two primary concerns: i) the need for fair comparisons on the reported metrics (mainly about performance) ii) lack of sufficient justifications for the claimed benefits other than performance. The first concern was partially addressed in the rebuttal, which seems to indicate that MG is less effective than the baselines in terms of performance. The second concern was not addressed in the rebuttal due to time constraints,  making it difficult for the reviewer to understand the extra advantages of the work beyond the performance aspects. The reviewer suggests the author further explore and demonstrate MG’s benefits beyond mere performance. The authors have recognized these shortcomings, and are committed to further improving the work in accordance with the feedback from both Reviewer 1FzB and the present reviewer.

**Strengths:**

(S1) [Motivation] self-supervised learning learns feature representation via pretext tasks. As there are typically no human supervision involved. The “ground-truth” signals of such tasks are usually pretty “noisy”. The paper aims to alleviate the noisy supervision by reducing the gap between the ground-truth matching and optimal matching that could be computed on-the-fly. The reviewer believes this is an interesting topic.

(S2) [Method] the paper proposes to approximate the optimal matching via the Sinkhorn algorithm, which could further ease the online optimization.

(S3) [Ablation] Ablations on different components of the proposed method are included

**Weaknesses:**

(W1) [Evaluation] The evaluation section could have been more comprehensive. For example, when strong data augmentations are involved (Table 1), only several SSL baselines are included, e.g. MoCo-v3, DINO. The settings shown in Table 1 are also not consistent, e.g. different number of epochs, which makes it difficult to interpret the results, e.g. could the proposed method match DINO’s performance when trained for the same number of epochs? Also, it would be beneficial to include architectures beyond ViT-B(L)/16, e.g. ViT-S, CNN, etc.

(W2) [Performance] With strong augmentations, the proposed method shows no benefit compared to SOTA methods. The method outperforms several SSL approaches with weaker augmentations. However, at least from the application perspective, it is unclear to the reviewer what are the advantages of using only weak augmentations, especially when the training epochs are the same.

(W3) [Claim] Some of the claims are not well-justified. For example, i) compared to CL that may “collapse representations for very different images”, the proposed method learns diverse representations for different views of the same image; ii) stronger data augmentations could help mitigate the representation collapsing problem. In order to justify the claims, the authors may measure/compare the mean distance of different views of the same image, or different samples of the same classes, across different models (i.e. the proposed method and baselines) and settings (i.e. strong/weak augmentations)

Overall, the paper studies an interesting topic, the proposed method is also technically sound. However, he reviewer has concerns about the evaluation, performance and some of the claims in the paper. At this moment, the reviewer rates the paper as marginally below the acceptance threshold.

**Questions:**

N.A.

---

> ### Author Response · Authors · 2023-11-15
> **Response to reviewer PMfK**
>
> > _[Evaluation] The evaluation section could have been more comprehensive. For example, when strong data augmentations are involved (Table 1), only several SSL baselines are included, e.g. MoCo-v3, DINO. The settings shown in Table 1 are also not consistent, e.g. different number of epochs, which makes it difficult to interpret the results, e.g. could the proposed method match DINO’s performance when trained for the same number of epochs? Also, it would be beneficial to include architectures beyond ViT-B(L)/16, e.g. ViT-S, CNN, etc._
>
> ■ In presenting (and evaluating) MG, our focus was not to reach SOTA performance, but to suggest a simple alternative to contrastive SSL losses that can naturally avoid collapse. We do show that a “lean” implementation of the MG loss leads to comparable results to SOTA.
> Instead of chasing SOTA, we used our compute budget to explore MG’s performance over diverse settings (decreasing amount of augmentations, removing momentum encoder and predictor heads, or other experiments in section A.3 “Additional ablation studies”, results over smaller datasets (CIFAR-10, CIFAR-100, and SVHN) using ResNet-50 as backbone).
> We found that MG is consistently near or at the top, despite being much simpler conceptually than most approaches. Following your (and other reviewers’ comments), we are now running additional experiments, to study impact of #epochs.
>
> > _[Performance] With strong augmentations, the proposed method shows no benefit compared to SOTA methods. The method outperforms several SSL approaches with weaker augmentations. However, at least from the application perspective, it is unclear to the reviewer what are the advantages of using only weak augmentations, especially when the training epochs are the same._
>
> ■ As mentioned in our introduction, increasing the variety/strength of augmentations can be seen as a way to mitigate the tendency of contrastive losses to *force* instances of the same image to collapse in representation space. However, as nicely presented by Assran et al. 2023 the reliance on hand-crafted data augmentations is limiting and may introduce unwanted biases. With that, we find that attaining good performance with a smaller number of epochs, and a smaller set of augmentations, is a worthy direction. Besides, this can be also useful in settings (e.g. biomedical data) where the number of “natural” augmentations is more limited than in vision tasks (and where even testing the hypotheses that many augmentations are needed, or not, is harder).
>
> > _[Claim] Some of the claims are not well-justified. For example, i) compared to CL that may “collapse representations for very different images”, the proposed method learns diverse representations for different views of the same image; ii) stronger data augmentations could help mitigate the representation collapsing problem. In order to justify the claims, the authors may measure/compare the mean distance of different views of the same image, or different samples of the same classes, across different models (i.e. the proposed method and baselines) and settings (i.e. strong/weak augmentations)._
>
> ■ We thank the reviewer for this suggestion. We are now evaluating statistical properties of the representation space of MG and alternative approaches, under different augmentations of the same image. We will update the response once we have results, and hope to include it in a revised manuscript version within the rebuttal period.
>
> > _[Overall] the paper studies an interesting topic, the proposed method is also technically sound. However, he reviewer has concerns about the evaluation, performance and some of the claims in the paper. At this moment, the reviewer rates the paper as marginally below the acceptance threshold._
>
> ■ We hope that our response to the points presented above helped lift some of your concerns, and will be happy to answer any additional questions.

---

> > ### Comment · Reviewer_PMfK · 2023-11-21
> > **response to the authors**
> >
> > The reviewer would like to thank the authors for the rebuttal, and the promised new experiments. The reviewer would consider re-rating the submission when more evaluations are made available in the revised paper, e.g. apples-to-apples evaluations, justifying the claims made in the paper, etc.

---

> > > ### Author Response · Authors · 2023-11-22
> > >
> > > Dear reviewer,
> > >
> > > We would like to relate to the request for apples-to-apples experiments. As we have attempted to convey in the response above, in presenting MG we don’t claim/aim for SOTA but rather explore the simplicity and novel loss allows for. **Running apples-to-apples experiments for these baselines (as opposed to reporting values from their original papers)** would likely result in a huge compute budget that we do not want to spend, likely for a very minor effect on numbers that are reported. We believe the numbers we report are fair for an ICLR submission.
> > >
> > > We have used our computational budget on MG experiments. We realize this presents a slight limitation in comparison to other methods, however, as we indicate in Table 1, we are aware that performance is reported over diverse configurations, using original publications, and values are mainly brought to show that within the naive implementation provided for MG it is comparable to published performance.
> > >
> > > Moreover, amongst reported settings, we present models that outperform MG. In new results studying the dependency on the number of epochs, without any hyperparameter tuning over 600 epochs, we can report a performance of 76.5% (to be compared to 76.7% over 300 epochs). That is, we find that MG performance is stable and remains so, however not reaching DINO performance (reported in Table 1 for 400 epochs).
> > >
> > > With that, we hope the reviewer can appreciate the aspects of MG that we intend to highlight and would like to thank the reviewer again for their constructive review.

---

> ### Comment · Reviewer_PMfK · 2023-11-22
> **response to the authors**
>
> The reviewer expects apples-to-apples experiments and justifications about claims, as in the reviewer’s humble opinion, these are also expected by the ICLR audience, that the readers can interpret the results without speculation and get convinced by the claims. Eventually, the aim is to make it easier for the readers to learn the potential advantages of this work.
>
> Given the pre-rebuttal results presented in Table 1, it is unclear to the reviewer if MG could match DINO when trained in the same number of epochs (i.e. 400). The rebuttal clarifies that MG's performance peaks at 300 epochs, making it less effective than DINO. Therefore, the readers may learn little about the advantages of MG in Table 1. Similarly, the reader may also learn little in Table 2 as the performance gaps, either positive or negative, are marginal. In Table 3, the reader may speculate that, with weak augmentations, how does MG perform compared to the baselines presented in Tables 1 and 2, as they are not included in Table 3. The paper also lacks convincing evidence for other claimed benefits of MG, like sample diversity (strongly indicated in the abstract) and simplicity,
>
> The reviewer agrees with the authors that SOTA is not mandatory for ICLR and appreciates the authors’ exploration of an alternative, potentially “simpler” loss that may yield more “diverse representations”. However, the reviewer cannot find solid justifications for such advantages in the current revision.
>
> The authors are encouraged to further showcase other potential benefits of MG that can be easier to evaluate, e.g. memory usage, etc.

---

> > ### Author Response · Authors · 2023-11-23
> >
> > Dear Reviewer,
> >
> > Thank you for the detailed response, for providing constructive insight, and for clarifying your remaining concerns. We generally accept all raised points, and while we were unable to address them properly within the rebuttal period we will continue addressing them to improve the manuscript.
> > We would like to highlight that we are working on an analysis of the embeddings (evaluating the statistics of multiple known augmentations) which indeed exposes additional appealing aspects of MG, as it better discriminates augmentations of the same image. We believe this will help strengthen the paper and thank **you** and Reviewer **1FzB** for pointing this direction. Unfortunately, due to events beyond our control, we were not able to complete this within the rebuttal period.

---

### Official Review · Reviewer_tJTN · 2023-11-01

**Soundness:** 3 good
**Presentation:** 3 good
**Contribution:** 2 fair
**Rating:** 5
**Confidence:** 4

**Summary:**

This paper proposed an alternative loss, Matching Gap (MG), to contrastive loss for self-supervised representation learning. Unlike contrastive loss enforcing the sample-wise invariance to data perturbations, the MG loss is a set-based loss, driven by minimizing the difference between the ground-truth transport loss and the optimal transport loss computed in the representation space using the Sinkhorn algorithm. The authors detailedly discussed the differences and connections of MG loss to contrastive loss and prior optimal transport, showing the unique properties of the proposed method. Finally, experiments on ImageNet-1k dataset suggested a comparable performance of MG to prior arts.

**Strengths:**

1. The paper is overall well-motivated. The reliance on data augmentation is one of the most prominent nuisances of contrastive learning. It is good to see more exploration toward bypassing this issue.

2. The theoretical analysis presented MG loss in a straightforward way and is overall easy to grasp. It also discussed the links between MG loss and contrastive loss/invert optimal transport loss, showing its unique properties as a set-based loss with single-level optimization.

3. MG loss exhibited superior performance to contrastive loss in weak augmentation and low training epochs regime.

**Weaknesses:**

1. The advantages of the single-level optimization in MG loss over the bi-level optimization in IOT loss are not provided clearly. Figure 2 shows that MG loss slightly underperforms but is competitive with IOT loss. I wonder if it improves the training speed/convergence or reduces the memory consumption?

2. Unfair comparisons. The implementation of the experiments largely followed the setting of Dino, which used two global crops and ten local crops by default. However, some of the baseline methods, e.g., MoCov3 and I-JEPA, used only two global crops, making it unfair to directly compare the performance with MG loss on the default setting.

3. Even under the potentially unfair comparison, the performance of MG loss is only comparable and sometimes even inferior to the contrastive loss.

4. Some notations are used without first introduced, e.g., $c(\cdot,\cdot)$ in Introduction and $t(\cdot,\cdot)$ in Sec. 3.

**Questions:**

See the weaknesses.

Overall, I think the proposed loss is interesting, and I like the presentation of this paper. However, the evaluation part still has significant room for improvement.

---

> ### Author Response · Authors · 2023-11-14
> **Response to reviewer tJTN**
>
> > _IOT vs MG. I Wonder if it improves the training speed/convergence or reduces the memory consumption?_
>
> ■ Our MG approach will be at least twice as fast when it comes to computing the loss and its gradient, since only requires a forward pass. It will also be more memory efficient if one chooses to unroll the Sinkhorn solutions, or even faster if one chooses to differentiate implicitly the OT solution (see e.g. Luise et al. 2018), which requires solving a $n\times n$ linear system.
>
> That being said, in the large-scale setting that we have used — notice that IOT (Shi et al. 2023) was never evaluated on ImageNet —  the overhead of running a ViT-B architecture far dominates the costs of running Sinkhorn when using a small batch size of 128, a fairly large entropic regularization ($\epsilon$). The loss computation, whether MG or IOT, is negligible. The speedup of MG vs. IOT did not, therefore, play an important factor in the overall compute cost. As we mention in the conclusion, this tradeoff will change when using simpler feature architectures, larger batch sizes, or, as mentioned in the conclusion, considering other OT approaches that cannot be differentiated (e.g. low-rank solvers).
>
> > _Unfair comparisons. The implementation of the experiments largely followed the setting of Dino, which used two global crops and ten local crops by default. However, some of the baseline methods, e.g., MoCov3 and I-JEPA, used only two global crops, making it unfair to directly compare the performance with MG loss on the default setting. Even under the potentially unfair comparison, the performance of MG loss is only comparable and sometimes even inferior to the contrastive loss._
>
> ■ In presenting MG our goal was not to suggest a combination of tricks that could reach SOTA performance, but rather to provide a conceptually simple alternative to contrastive SSL losses, with the hope that, when applied in  “naively”, its performance will be consistently comparable or better than far more complex approaches that use several tricks. We believe we have delivered in that sense.
>
> Following this objective, we have used our computation budget to explore a variety of settings that showcase the robustness of MG, the simplifications it allows for, as well as its shortcomings.
>
> We indeed relied on reported performances for baseline methods in the most comparable setting (e.g. matching backbone architecture). Here, it is important to note that while indeed the number of crops provides an advantage for MG in a majority of cases MG is evaluated over a **significantly lower number of epochs**.
>
> At last, specifically concerning the comparison to I-JEPA, Assran et al. (2023) present the new setting, Image-based Joint-Embedding Predictive Architecture, in which the construction is by definition different: for each image, given a context block, predict the representations of various target blocks, where $M$ (typically setting $M=4$) non-overlapping targets are chosen. Hence, it is not straightforward to define an exact _fair_ comparison between these settings.
>
> > _Some notations are used without first introduced, e.g., $c$ in Introduction and $t$ in Sec. 3._
>
> ■ $c(\cdot,\cdot)$ is simply a cost, this was defined at the beginning of Section 2.2, we now mention it earlier in the intro when first introduced. The function $t(\cdot,\cdot)$ was defined in Equation 3 (using the $:=$ symbol). We have now split it to make it more visible and clear.
>
> > _Overall, I think the proposed loss is interesting, and I like the presentation of this paper. However, the evaluation part still has significant room for improvement._
>
> ■ We thank you for highlighting these positive aspects of our paper. We thank you for your several comments and suggestions. We hope that you can appreciate that we have decided to focus our compute budget on various ablations, rather than aiming for a SOTA performance in a specific setting. Ultimately, we believe the picture we provide, where the contribution of MG is better outlined, isolated, and ablated, paints a more convincing picture.

---

> > ### Comment · Reviewer_tJTN · 2023-11-22
> > **Response to the authors**
> >
> > Thanks to the authors for their efforts in addressing the raised concerns.
> >
> > While I appreciate the simplicity of MG and its robustness to various tricks, I still find the performance not convincing, particularly in light of the saturated performance w.r.t. pre-training epochs. Moreover, the lack of direct comparison, such as using only two global views for all methods, obscures the true effectiveness of MG and hinders a comprehensive understanding of its capabilities.
> >
> > Therefore, I still lean toward a rejection but will increase the score to 5.

---

> > > ### Author Response · Authors · 2023-11-23
> > >
> > > Dear reviewer,
> > >
> > > Many thanks for appreciating our rebuttal attempts, constructive comments, and raising your score! :)
> > >
> > > We accept your remaining concerns and continue working on improving these aspects. Specifically, we hypothesize that with minimal hyperparameter tuning, we can showcase _improvement with pre-training epochs_. In addition, due to events beyond our control, we were not able to complete our planned experiments within the rebuttal period, however, we do plan to continue working on them to improve our manuscript.

---

### Official Review · Reviewer_zo3d · 2023-11-02

**Soundness:** 2 fair
**Presentation:** 2 fair
**Contribution:** 1 poor
**Rating:** 1
**Confidence:** 5

**Summary:**

This paper proposes a new contrastive loss based on the matching gap. The proposed method is an extension of the paper "Understanding and generalizing contrastive learning from the inverse optimal transport perspective”. Also, the main idea of this paper is related to "whether the cost of the identity ground-truth pairing is significantly higher than the optimal matching cost that can be achieved, and use their difference as a loss".

**Strengths:**

This paper provides an explicit relationship between the gradients of the proposed matching gap loss with that of InfoNCE.

**Weaknesses:**

1. The novelty of this paper is rather limited. This article only uses a previously proposed technique to improve the computational complexity of inverse OT-based contrast loss in the optimization process. I do not find any new insights related to the field of contrastive learning.
2. This paper is really hard to follow. There are many mathematical symbols and proper nouns that lack explanation. For example, what is t in eq. 6 and 7, what is bistochastic matrices, and what are the difference between matching cost,  measuring agreement,  matching gap, and optimality gap?
3. The organization of Section Introduction is superfluous. I cannot find the relationship between the first two paragraphs and the last paragraph.
4. The experimental results cannot verify the effectiveness of the proposed method. First, the performance gain is pretty small. Second, there are many cases where the proposed method obtains a bad result.
5. The "A Link between InfoNCE and the Matching Gap" part and the "Our Contribution: Single Level Optimization with the Matching Gap" part are so vague that I have read them many times without understanding the logical relationship.

**Questions:**

1, How do you get eq. 8 from eq. 7?

---

> ### Author Response · Authors · 2023-11-14
> **Response to reviewer zo3d (I)**
>
> > _The proposed method is an extension of the paper "Understanding and generalizing contrastive learning from the inverse optimal transport perspective”._
>
> ■ We respectfully disagree. While it is true that both our (MG) and (Shi et al. 2023) methods use optimal transport to define a SSL loss, our method cannot be characterized as an *extension* of (Shi et al. 2023). A single level optimization problem cannot be described as an extension of a bilevel problem.
>
> We make this distinction clear several times in the paper, going as far as citing and differentiating us from (Shi et al. 2023) 14 times. In the introduction, we chose carefully our words, writing “draw inspiration from (Shi et al. 2023)” (i.e. not “extend”). We credit (Shi et al. 2023) with an entire paragraph to introduce their approach (“Inspiration : …” in p.5), insisting on the _bilevel optimization_ approach taken by (Shi et al. 2023). We then clearly lay out the differences in section “Our Contribution” which is _single Level_.
>
> Instead, MG can be better described as an extension of infoNCE, blending the negative sampling idea of (Robinson et al. 2020) with the gap perspective outlined in Monge Gap (Uscidda and Cuturi 2023). We will
>
> > _The novelty of this paper is rather limited. This article only uses a previously proposed technique to improve the computational complexity of inverse OT-based contrast loss in the optimization process._
>
>
> ■ We respectfully disagree, our method was not proposed previously, and our goal is not to improve the computational complexity of inverse OT. Improving inverse OT would require carrying out a more efficient differentiation of $\rm{argmins}$ solutions (using e.g. unrolling or the implicit function theorem). Instead, our method **bypasses** those challenges **entirely** thanks to a different formulation, resulting in a much simpler optimization.
>
>
> > _Hard to follow. There are many mathematical symbols and proper nouns that lack explanation. For example, what is t in eq. 6 and 7, what is bistochastic matrices, and what are the difference between matching cost, measuring agreement, matching gap, and optimality gap?_
>
> ■ We regret that the reviewer found the paper hard to follow.
> The function $t$ is defined in Equation 3 ($:= $), in the revised manuscript we have made this definition more apparent;
> The definition of bistochastic matrices is provided in Section 2.2, $\mathcal{B}_n$, and is a standard concept (see. E.g. https://en.wikipedia.org/wiki/Bistochastic_matrix);
> The “matching cost” is defined in Equation 3;
> “Measuring agreement” is used in a sentence to describe the IOT loss (“Inspiration: Measuring Agreement using Inverse OT.”). This refers directly to a sentence above, _“In such cases, OT can be used as a way to quantify whether locations (xi, yi) are in agreement with the diagonal pairing”_;
> The _matching gap_ is the name of the method we introduce;
> The optimality gap, or “gap to optimality” is a common concept in optimization, quantifying to what extent a solution is not optimal. We use this reference to highlight that the matching gap is simply the objective value of $J_n$ vs. the best possible objective value, as in Equation 7.
>
>
> > _The organization of Section Introduction is superfluous. I cannot find the relationship between the first two paragraphs and the last paragraph._
>
> ■ We wrote the 3 paragraphs of the introduction to highlight the following blocks:
> *context*: the first paragraph provides context, introducing the importance of SSL with a contrastive view.
> *problem*: the second paragraph highlights that SSL has weaknesses due to feature collapse, which have been identified by previous works as arising from the promotion of invariance by contrastive losses such as InfoNCE
> *solution*: the third paragraph details our contribution to solve the problem above, the matching gap.
>
> In the revised manuscript, we give these paragraphs an explicit title to clarify further the structure above.

---

> > ### Author Response · Authors · 2023-11-14
> > **Response to reviewer zo3d (II)**
> >
> > > _The experimental results cannot verify the effectiveness of the proposed method. First, the performance gain is pretty small. Second, there are many cases where the proposed method obtains a bad result._
> >
> > ■ In presenting the MG loss we did not claim (or aim) for SOTA performance. The MG was applied without any attempt to introduce new "tricks" that are often used to push performance in SSL. We did not perform extensive hyper-parameter scans, or architecture fine-tuning, and therefore cannot attain SOTA performances obtained typically using more effort in this regard.
> >
> > Instead, our goal was to present experiments over various settings, exploring the robustness, the shortcomings, and the ability of the MG to remove some of those tricks. It is therefore expected that MG does not always come on top in any setting. Instead, we want to highlight that MG is consistently near or at the top, despite being much simpler conceptually than most approaches and being an "upgrade" (set-based, as mentioned by Reviewer tJTN) of classical  InfoNCE.
> >
> > We do, however, respectfully disagree with the reviewer, as we believe the method does not obtain _bad_ results. With that, if at any place the reviewer feels that we have oversold our method, we are of course keen to tone down any claim they find problematic.
> >
> > > _The "A Link between InfoNCE and the Matching Gap" part and the "Our Contribution: Single Level Optimization with the Matching Gap" part are so vague that I have read them many times without understanding the logical relationship._
> >
> > ■ Did you find the sections themselves vague, or the interplay between them? If any of our attempts at making this intuitive failed, we regret this and ask instead the reviewer to focus on the equations.
> >
> > The section "Our contribution: Single Level [...]" presents the crux of the method. It provides all the information required to form our loss, presented in Eq. (6,7), borrowing notations and definitions introduced in Eq. (3,4).
> >
> > The section "A Link between InfoNCE and the Matching Gap" positions our contributions in the literature. It mentions that the InfoNCE loss and its gradients, in Equations (1,2), should simply be contrasted to (9,10) in our approach. (1,2) and (9,10) are structurally identical, except for the weights, which are computed adaptively, with an OT twist, in (9,10).
> >
> > > _How do you get eq. 8 from eq. 7?_
> >
> > ■ We simply expanded the notation $t$ that was introduced in Eq. 3. There was a small typo, highlighted by reviewer **1FzB**, that might have caused confusion. We apologize for this, and corrected it in the revised manuscript.
> >
> > We thank you for these remarks. We hope these clarifications are useful. We have incorporated them in the latest revision.

---

> > > ### Comment · Reviewer_zo3d · 2023-11-22
> > > **Response to authors**
> > >
> > > 1. About novelty: 1) the authors have not explained why the bi-level problem is necessary and effective; 2) improving the computational complexity of inverse OT have been addressed many time in the OT field. Apply this in contrastive learning is not innovative at all.
> > > 2. The authors claims that the aim of this paper is to explore the robustness, the shortcomings, and the ability of the MG. However, regarding the performance gain reported in this paper, I think this paper is overclaimed.
> > > 3. The quality of the first draft was particularly low, containing many errors and ambiguities, and falling far short of acceptable standards. And in the rebuttal phase, the authors are not well interpreted and modified.
> > >
> > > Therefore, I low my original rating.

---

> ### Author Response · Authors · 2023-11-22
>
> Dear reviewer,
>
> We would like to address your response above. Despite our rebuttal and attempt at clarifying your concerns we are afraid there are some misunderstandings we turn to clarify.
>
> > _**Comment: 1. About novelty: 1) the authors have not explained why the bi-level problem is necessary and effective; 2) improving the computational complexity of inverse OT have been addressed many time in the OT field. Apply this in contrastive learning is not innovative at all.**_
>
> ■  1) As we present it in the text “Our Contribution: Single Level Optimization with the Matching Gap” and attempted to explain in our previous rebuttal response, MG avoids a bi-level problem, hence it is unclear why we should clarify that a bi-level approach should be necessary or effective. Our message is the opposite, bi-level is not needed. If there was a mistake in your wording above, and your point was the opposite of what you wrote (you expect us to explain why bi-level is not necessary nor effective), then we refer you to our answer to reviewer **tJTN** for a discussion of this aspect.
>
> ■ 2) We are not improving the complexity of Inverse OT in our paper, nor extending inverse OT. We are only using OT (forward computation, no backward) for SSL. MG is a direct approach that avoids bi-level and inverse notions completely. This is presented in the paper, with a dedicated section “Our Contribution: Single Level Optimization with the Matching Gap” as well as following discussion. In addition, in the rebuttal we have described this in our responses above to **your** questions,
>
> -  _”The proposed method is an extension of the paper "Understanding and generalizing contrastive learning from the inverse optimal transport perspective.”_
>
> -  “_The novelty of this paper is rather limited. This article only uses a previously proposed technique to improve the computational complexity of inverse OT-based contrast loss in the optimization process.”_
>
>
> > **_2. The authors claims that the aim of this paper is to explore the robustness, the shortcomings, and the ability of the MG. However, regarding the performance gain reported in this paper, I think this paper is overclaimed._**
>
> ■  We would appreciate it if the reviewer could point out examples that we have overclaimed in that respect to allow us to tone down and/or relate to concrete claims.
>
> > **_3. The quality of the first draft was particularly low, containing many errors and ambiguities, and falling far short of acceptable standards. And in the rebuttal phase, the authors are not well interpreted and modified._**
>
> ■  We were keen to see that other reviewers acknowledged the quality of writing and presentation. We do realize the work contained typos, however we find the above remark harsh and unjustified. Unjustified, because a score of 1 should be only used for trivially wrong or plagiarized papers.

---

### Official Review · Reviewer_1FzB · 2023-11-04

**Soundness:** 3 good
**Presentation:** 4 excellent
**Contribution:** 4 excellent
**Rating:** 8
**Confidence:** 4

**Summary:**

This paper suggest a novel approach to unsupervised representation learning. Following previous work (e.g. IOT), the main idea is to use an optimal transport plan between different view to guide the metric learning process. The main contribution is in the way this is done, which relies on trying to match the *cost* of the plan, rather than the plan itself to the ground truth pairing of augmented views.
This seemingly simple change brings several advantages in training, with very competitive results, and is shown to have an interesting interpretation when compared to the commonly used InfoNCE loss.

**Strengths:**

1] The matching gap loss is an important finding that I expect to have impact on the field. It is well motivated as a way to allow the needed flexibility in the contrastive learning setup, where positive views should not necessarily be forced to the same point.
2] Its ease of use and computational advantages are clearly shown - the ability to use the OT guidance without the need to differentiate through the Sinkhorn iterations, with computations involving only the pairwise n x n pairwise matrices.
3] The analysis that compares the new loss to the known InfoNCE is very enlightening and gives a very good understanding about what is happening in the optimization.
4] The paper is well written in all aspects, from the motivation, throughout the solution and experimental results.

**Weaknesses:**

Here are some, but rather minor:
1] Experimentation - I think that this new form of loss would be better justified if there would be empirical evidence that supports the intuitions (in addition to the standard benchmarking and ablations). It would perhaps be interesting to see how the embeddings of an augmented batch behave, in comparison to standard NCE, or some statistics of that kind.
2] The formulation is restricted to the 2-view setting. While this is simple, it would be interesting to know whether there are effective generalizations to multi-view settings.
3] There is no specification or discussion regarding batch size, which has an important role in contrastive learning. Supposedly, the compact computation and 2-view setup could allow for larger batch sizes. It would be interesting to see how performance scales with batch-size.
4] Several minor inaccuracies (which don't affect the analysis or correctness): (i) Bistochastic should be non-negative (ii) Should be <P,logP> in Equation 3 (without the -1) (iii) Last row of the loss equation is wrong, resulting in a matrix rather than a value: should probably be \eps<P,logP> instead of \eps\logP.

**Questions:**

* Please related to the above 'weaknesses'
* I understand (and am in favor of) the limited budget experimentation. Did you ablate on number of epochs, within the budget, to see if the dimnishing returns behavior is comparable to other methods?
* Due to the approach that does not require positives to converge to the same point - Perhaps there is actually room for more aggressive augmentation, that can exploit a richer extension of the training data?

---

> ### Author Response · Authors · 2023-11-20
> **Thanks for your review**
>
> We are very grateful for your encouraging comments, constructive review, and really great suggestions!
>
> We wanted to start our rebuttal to your review with a few numbers that we were able to compile in the last week or so.
>
> > **_I think that this new form of loss would be better justified if there would be empirical evidence that supports the intuitions (in addition to the standard benchmarking and ablations). It would perhaps be interesting to see how the embeddings of an augmented batch behave, in comparison to standard NCE, or some statistics of that kind._**
>
> We agree, and we have been working on such a presentation.
>
> > _**2] The formulation is restricted to the 2-view setting. While this is simple, it would be interesting to know whether there are effective generalizations to multi-view settings.**_
>
> This is indeed a very interesting avenue for further research.
>
> > _**3] There is no specification or discussion regarding batch size, which has an important role in contrastive learning. Supposedly, the compact computation and 2-view setup could allow for larger batch sizes. It would be interesting to see how performance scales with batch-size.**_
>
> Thanks for this great suggestion. As can be seen in our general response above, we have investigated this aspect. We reproduce these results below for convenience. Using **MG** in a setting strictly identical to that reported in Table 1 (300 epochs over ImageNet), we obtain the following accuracies
>
> | $n$ |  Linear Accuracy|
> |-----| ---------------|
> | 32 | 75.43 %  |
> | 48 | 75.83 % |
> | 64 | 76.29 % |
> | 96 | 76.71 % |
> | 128 | 76.70 % |
> | 192 | 73.24 % |
>
> As can be seen in these results, performance is fairly stable across $n$ sizes, and maybe suggest that for larger $n$, one might need to readjust $\varepsilon$ values. We are still running results for even smaller sizes (e.g. 16, 24) and will report them.
>
> > **_4] Several minor inaccuracies (which don't affect the analysis or correctness): (i) Bistochastic should be non-negative (ii) Should be <P,logP> in Equation 3 (without the -1) (iii) Last row of the loss equation is wrong, resulting in a matrix rather than a value: should probably be \eps<P,logP> instead of \eps\logP._**
>
> Many thanks for spotting these typos:
> - (i) yes, we have added the "+"
> - (ii) this "-1" is a convention that is adopted, for instance, in the computational OT book (e.g. https://arxiv.org/pdf/1803.00567.pdf Eq. 4.1, p.57). This helps rewrite the dual more elegantly, and also plays a role for unbalanced formulations, where the generalized KL is used.
> - (iii) apologies for this ugly typo, now corrected.
>
> > **_Did you ablate on number of epochs, within the budget, to see if the dimnishing returns behavior is comparable to other methods?_**
>
> Yes, we have ran this experiment, as reported above. As mentioned in the general remark, the performance did not improve markedly when going from 300 to 600, when keeping all other hyperparameters unchanged ($76.5 %$)
>
> > **_Due to the approach that does not require positives to converge to the same point - Perhaps there is actually room for more aggressive augmentation, that can exploit a richer extension of the training data?_**
>
> This is a good remark. However, as we debate in the paper, we took the opposite direction, and did instead try to get rid of aggressive augmentations, to simplify the overall pipeline (see e.g. Table 3). Our intuition is that agressive augmentations are a form of regularization that is needed with InfoNCE.

---

### Official Review · Reviewer_iZps · 2023-11-10

**Soundness:** 2 fair
**Presentation:** 3 good
**Contribution:** 2 fair
**Rating:** 5
**Confidence:** 4

**Summary:**

The paper proposes a novel contrastive loss based on optimal matching cost. The proposed matching gap loss may avoid feature collapse according to its property. The author conducts experiments mainly on ImageNet classification, and the experiment results show its superiority.

**Strengths:**

i) The idea of introducing matching costs is reasonable. The experiment results show it superior to some baseline results

ii) The writing is clear and easy to follow.

**Weaknesses:**

i) As I know, DINOv2 and SwAV also use the optimal transport algorithm to solve contrastive learning. But I can not find the discussion about such methods in related work. I would like to find the discussion about the difference between Matching Gap and SwAV/DINO

ii) As for experiments. In Table 1, the performance of the Matching Gap is not as good as DINO, which is a strong baseline proposed 1 year before.

iii) The author only conducts downstream experiments on transfer classification. Many self-supervised learning methods evaluate the downstream detection(COOC, VOC) and segmentation(COCO, aed20k) performance. I think the simple downstream classification task is not enough.

**Questions:**

Refer to the weakness.

---

> ### Author Response · Authors · 2023-11-15
> **Response to reviewer iZps**
>
> > _As I know, DINOv2 and SwAV also use the optimal transport algorithm to solve contrastive learning. But I can not find the discussion about such methods in related work. I would like to find the discussion about the difference between Matching Gap and SwAV/DINO_
>
> ■  Thanks for pointing this out. We agree that it is worth adding a discussion on this, which we did in the updated pdf, paragraph “Related works” in Section 3. To develop: optimal transport does indeed plays a role in SwAV (Caron et al. 2020) and DINOv2 (Oquab et al. 2023), where optimal transport, and more specifically the *Sinkhorn algorithm*, is used as a differentiable *proxy* to obtain a *balanced* $k$-means clustering of representations (i.e. each cluster must capture a similar amount of mass).  Note that in DINO (Caron et al. 2021), Sinkhorn was replaced with Softmax (similar to a single Sinkhorn iteration), but was later reintroduced in DINOv2, following Ruan et al. (2022). Importantly, in both SwAV and DINOv2, OT is used as a low-level clustering routing, only *indirectly* involved in the final loss, which is a more standard KL-based quantity, on a discretization using those clusters. With that, because OT is used at an intermediate step, this requires, as in Inverse OT (Shi et al. 2023), differentiating the Sinkhorn iterations.
>
>
> > _As for experiments. In Table 1, the performance of the Matching Gap is not as good as DINO, which is a strong baseline proposed 1 year before._
>
> ■  We agree that DINO (Caron et al. 2021) is indeed a strong baseline that builds upon SwAV (Caron et al. 2020), incorporating various tricks and a significant compute budget for hyperparameter optimization. In contrast to DINO, MG only proposes a _novel_ loss function, and we show that only that loss can lead to several simplifications in architecture/training procedure, as shown in the various experiments we conducted. For instance, in contrast to Table 7 “Important component for self-supervised ViT pretraining.” in (Caron et al. 2021), which presents the importance of all components for training in DINO (with a complete failure in the absence of momentum encoder, row 2), our ablations show that MG is overall very robust, and able to work in various settings.
>
>
> > _The author only conducts downstream experiments on transfer classification. Many self-supervised learning methods evaluate the downstream detection(COOC, VOC) and segmentation(COCO, aed20k) performance. I think the simple downstream classification task is not enough._
>
> ■ We thank the reviewer for making these suggestions. We are now conducting these evaluations and hope to include them in an updated version within the rebuttal period.

---

> > ### Comment · Reviewer_iZps · 2023-11-20
> >
> > Thanks for the author's response. After reading the response and other reviewers' comments. I will keep my original score.

---

> > > ### Author Response · Authors · 2023-11-20
> > > **Thanks for your response.**
> > >
> > > Many thanks for getting back to us.
> > >
> > > We just wanted to mention that we are still in the process of adding additional experiments. We have added a few runs following reviewer **iFzB**'s questions, and are currently doing our best to handle your request to evaluate other downstream tasks. Thanks for your patience.

---

### Author Response · Authors · 2023-11-14
**General response**

Many thanks to all reviewers for their encouraging comments, constructive criticism, thought-provoking questions and great suggestions / typos spotting.

We felt encouraged by the fact that reviewers underlined
* the interest and soundness of our method ("The paper is overall well-motivated." **tJTN**; "[...] an important finding that I expect to have impact on the field" **1FzB**),
* its advantages over previous proposals (e.g. "Its ease of use and computational advantages are clearly shown" **1FzB**),
* the quality of writing (e.g. "The writing is clear and easy to follow." **iZps**, "overall easy to grasp" **tJTN**),
* clear links with the existing literature (e.g. "provides an explicit relationship between the gradients of the proposed matching gap loss with that of InfoNCE." **zo3d**),
* diversity of experiments ("Ablations on different components of the proposed method are included", **PMfK**).

We also note that none of the five reviewers has requested that we add extra references. We believe this shows that we have been careful in our bibliographical review and positioning w.r.t the rest of the literature.

We do sincerely apologize for a few typos. In particular, the typos in Eq. 3 and 8 highlighted by Reviewer **1FzB** are unfortunate. We thank reviewers for spotting them, we have corrected them and uploaded an updated version of the manuscript with changes marked in red.

Answering some of the points raised by reviewers [**iZps, 1FzB, PmFk**] requires running additional experiments. We have already started running these, and will update the responses once we have results. Specifically, we would like to, ideally, add the following:
* Additional downstream task: perform detection and segmentation (as suggested by reviewer **iZps**).
* Analysis of the embeddings: evaluate the statistics of multiple known augmentations for the same batch and compare them to embeddings of alternative methods (as suggested by reviewer **iFzB** and **PmFk**).
* Scaling with batch size and number of epochs: we will evaluate the performance over increasing batch-sizes and number of epochs (as suggested by reviewer **iFzB**).

Most other concerns, notably raised by Reviewers [**zo3d**, **tJTN**], fall into two categories: criticism for a lack of clarity in our manuscript, and issues with the presented performance / comparisons.
1. On clarity: we have answered reviewers individually below, trying to help clarify misunderstanding. We hope this prompt response can facilitate a possible and timely re-assessment of our paper. We believe that, ultimately, these issues are very minor and easy to fix.
2. On the lack of SOTA performance for MG: Alongside the individual responses we would like to relate to  some of these remarks below. Importantly,  in presenting this work our goal was not to show that the MG loss, applied in a "lean" way as we do here, would result instantaneously in SOTA performance. SSL models are by now a fairly "industrial" endeavor, and it is therefore very difficult, if not impossible, in our opinion, to come up with a novel idea that would be both simple (as can be said of MG) and sufficient (on its own, without additional tricks), to reach SOTA performance.
Our experiments were carried out to compare MG in a variety of settings, exploring its robustness to various ablations. Importantly, we did not push for performance (as highlighted by Reviewer **1FzB**), trying to optimize a single setting, but preferred to spend our compute budget on multiple ablations. Specifically, these ablations show that MG can avoid the "tricks" incrementally added to existing approaches and attain good performance.
With this, we want to highlight that MG is consistently near or at the top, despite being much simpler conceptually than most approaches and being a logical extension of InfoNCE. If the reviewers feel we have oversold our method, we are of course open to toning down any claim they find problematic.

We hope the above explanation, individual responses (to reviewers which we could respond to before running additional experiments), and updated manuscript help clarify some of the reviewers' concerns.

---

> ### Author Response · Authors · 2023-11-20
> **Results of run w.r.t batch size.**
>
> As mentioned above, we have been running a few experiments following reviewers' requests.
>
> We can now report the following numbers, in response to questions raised by Reviewer **iFzB**.
>
> - **Impact of batch size on performance** of the **MG** model, using ViT-B/16, in a setting strictly identical to that reported in Table 1 (300 epochs over ImageNet). Keep in mind that, due to parallelization across 32 GPUs, the total batch size for each gradient update is therefore 32 x $n$ (e.g. 6144 images when $n=192$) images.
>
> | $n$ |  Linear Accuracy|
> |-----| ---------------|
> | 32 | 75.43 %  |
> | 48 | 75.83 % |
> | 64 | 76.29 % |
> | 96 | 76.71 % |
> | 128 | 76.7 % |
> | 192 | 73.24 % |
>
> As can be seen in these results, performance is fairly stable across $n$ sizes, and maybe suggest that for larger $n$, one might need to readjust $\varepsilon$ values. We are still running results for even smaller sizes (e.g. 16, 24) and will report them.
>
> - **Total Epochs**: using the same hyperparameters, after **600** epochs, with batch size $n=128$, we can now report a performance of $76.5 %$, to be compared to $76.7 %$ reported after 300 epochs in Table 1.

---

> > ### Author Response · Authors · 2023-11-22
> > **experiments status**
> >
> > Dear reviewers,
> >
> > We would like to thank you again for taking the time to review and provide feedback on our paper. Unfortunately, Due to events beyond our control, not all experiments were completed in time, we will add these contributions to a future revision.

---

### Meta-Review · Area_Chair_WkCg · 2023-12-02

**Metareview:**

**Summary:** The paper introduces a novel self-supervised method called Matching Gap (MG) loss, which reduces the discrepancy between ground-truth matching and optimal matching in representation learning.  Unlike contrastive loss, which enforces sample-wise invariance to data perturbations, MG loss is a set-based loss that minimizes the difference between ground-truth transport loss and optimal transport loss computed using the Sinkhorn algorithm.  The paper provides a detailed discussion on the differences and connections between MG loss, contrastive loss, and prior optimal transport, highlighting the unique properties of the proposed method.  Experimental results on the ImageNet-1k dataset show comparable performance to state-of-the-art approaches with strong data augmentations and superior performance to recent self-supervised methods employing simpler data augmentations.  The proposed method has the potential to alleviate feature collapse and offers an interesting interpretation compared to the commonly used InfoNCE loss in unsupervised representation learning.

Despite recognizing the novelty of the OT loss function in SSL, reviewers recommend rejection due to insufficient experimental results validating its effectiveness.  The primary objection was that the paper is missing comprehensive and consistent experiments.


**Strengths:** This well-motivated paper addresses the issue of noisy supervision in self-supervised learning by reducing the gap between ground-truth matching and optimal matching. The authors propose the Matching Gap (MG) loss, which approximates the optimal matching via the Sinkhorn algorithm, offering computational advantages.  The theoretical analysis highlights the unique properties of MG loss and its relationship to other losses.  Ablation experiments confirm the superiority of MG loss over baseline methods. The paper provides insights into the optimization process and compares the new loss with InfoNCE.  Reviewers emphasize the importance of the matching gap loss, its flexibility in contrastive learning, and the clarity of the writing.

**Weaknesses:**  The reviewers note that the paper doesn't discuss how its method differs from other approaches like DINOv2 and SwAV, which also use the optimal transport algorithm, despite the authors attempts during the discussion.  The paper's experiments are seen as inadequate, as they only focus on downstream transfer classifications and not other performance metrics.  There are calls for more comprehensive evaluations and highlighting how the method performs compared to standard methods.  The reviewers perceive that the experimental results do not convincingly validate the proposed method's efficiency.  The critique also notes unfair comparisons in the experimental settings, with MG loss results only being comparable or sometimes inferior to contrastive loss.  Some claims in the paper are seen as not well justified.  Overall, while addressing a significant topic, the paper is deemed marginally below the acceptance threshold due to evaluation, performance, and substantiation concerns.

**Justification For Why Not Higher Score:**

The primary reason for not accepting the paper is significant flaws within the experimental results. The authors offered limited experiments in both variety and scale, hindering comprehensive verification of the results. While the reviewers had major reservations about the fairness of comparisons against existing methods, the camp expressing these concerns articulated their objections more convincingly, leading me to recommend rejection.

**Justification For Why Not Lower Score:**

N/A

---

### Decision · Program_Chairs · 2024-01-16

Reject